# Molecular and Functional Characterization of α Chain of Interleukin-15 Receptor (*IL-15Rα*) in Orange-Spotted Grouper (*Epinephelus coioides*) in Response to *Vibrio harveyi* Challenge

**DOI:** 10.3390/ani13233641

**Published:** 2023-11-24

**Authors:** Yilin Zhang, Fan Wu, Guanjian Yang, Jichang Jian, Yishan Lu, Zhiwen Wang

**Affiliations:** 1Guangdong Provincial Engineering Research Center for Aquatic Animal Health Assessment, Shenzhen Public Service Platform for Evaluation of Marine Economic Animal Seedings, Shenzhen Institute of Guangdong Ocean University, Shenzhen 518120, China; 15342702503@163.com (Y.Z.); wfjprs@163.com (F.W.); yangguanjian2016@126.com (G.Y.); 2Guangdong Provincial Key Laboratory of Aquatic Animal Disease Control and Healthy Culture, Key Laboratory of Control for Diseases of Aquatic Economic Animals of Guangdong Higher Education Institutes, College of Fisheries, Guangdong Ocean University, Zhanjiang 524088, China; jianjc@gdou.edu.cn

**Keywords:** *IL15R*
*α*, *Vibrio harveyi*, orange-spotted grouper, HKLs, protein function

## Abstract

**Simple Summary:**

In recent years, grouper industry has been severely devastated by vibriosis. The present study explored the function of IL-15Rα (α Chain of Interleukin-15 Receptor) response to *Vibrio harveyi* infection from mRNA and protein level, respectively. We concluded that: a) at the mRNA level, IL-15Rα was significantly upregulated in each tissue after *V. harveyi* infection of grouper, as well as in HKL cells after stimulated by pathogenic stimuli. b) at the protein level, IL-15Rα can increase the cytokine expression level and promote the proliferation of HKL cells, and can also improve the survival of groupers after *V. harveyi* infection. Our study contributes to a better understanding of the immune responses of Ec-IL15Rα (*Epinephelus coioides* IL15Rα) in preventing *Vibrio* infection in groupers.

**Abstract:**

Interleukin-15 (IL15) is a proinflammatory cytokine that could induce the production of inflammatory cytokines. In this study, the α chain of the IL15 receptor of *Epinephelus coioides* (Ec-IL15Rα), a natural regulator of IL15, was identified, and immune response functions of fish were determined and characterized. Ec-IL15Rα contains a 720 bp open reading frame that encodes 239 amino acids, including four typical conserved cysteine residues with a highly conserved sushi domain. Ec-IL15Rα is closely related to *Epinephelus lanceolatus* and is the most clustered with teleost. Subcellular localization studies showed that Ec-IL15Rα was situated in the cytoplasm and cell membrane. *Ec-IL15Rα* was detected in 11 tissues, with the highest expression in the liver and blood. Meanwhile, the *Ec-IL15Rα* transcriptional levels substantially increased in nine tissues after *Vibrio harveyi* infection. *Ec-IL15Rα* was significantly up-regulated in HKLs by ConA, PHA, LPS and poly I:C stimulation. In vitro analysis, the recombinant protein of rEc-IL15Rα stimulates HKL proliferation and *IL1R*, *IL6R*, *IL10*, and *IL16* expression. Challenge experiments revealed that IL15Rα protein showed an increase of 6.67–10% survival protection rate after *V. harveyi* infection. This study provides a better understanding of the immune protection of IL15Rα in vertebrate fish.

## 1. Introduction

Cytokines are key mediators involved in regulating immune response. Interleukin-15 (IL15) is a pro-inflammatory cytokine produced mainly by activated monocytes, macrophages and dendritic cells, and it can activate immune cells [1]. The IL15 signal is transmitted through three receptors, including IL2, IL15RβγC and IL15Rα. IL15 has moderate affinity and binds to the IL2/IL15RβγC dimeric complex, and affinity binds to IL15Rα trimeric complex [2,3]. The 15Rα/15 complex is the most effective in promoting CD8^+^ memory T-cells and NK cells proliferation, but only when IL15 is bound exclusively with IL15Rα [4,5,6,7,8]. Trans-presentation is a mechanism of cytokine delivery that has been describe, that is IL15R and IL-15 encounter each other in the endoplasmic reticulum (ER) and are transported together to the cell surface where the cell surface complex can stimulate neighboring cells through the IL-15Rβ/γC [9,10]. IL15/IL15Rα complex could trigger signal transduction events in many types of non-immune and immune-related cells [6,7,8]. Then, their signal cascades induce various functions including apoptosis inhibition and proliferation in both immune and non-immune cells [11,12]. The IL-15 receptor shares many structural similarities with IL-2 receptors, including a consensus repeat called “sushi” domain which is thought to be critical for protein interaction [13]. Differently, compared to IL-2Rα, IL15Rα has a wider range of cellular targets, including T cells, monocytes/macrophages, NK cells etc. [1,14].

IL15Rα was first identified in human and mouse as a specific receptor for IL15. IL15Rα gene gaps of human and mouse were found in chromosome 10 and chromosome 2, with the encoding protein sharing 54% homology [13,15]. In teleost, IL15Rα has only been demonstrated in *Misgurnus anguillicaudatus* [16], *Oncorhynchus mykiss* [17] and *Oplegnathus fasciatus* [18], although several fish IL15Rα nucleotide and amino acid sequences could be checked in the GenBank database. Studies showed that IL15Rα could provide immune protection to fish against bacterial, virus and parasitic infection. After infection with *Edwardsiella tarda* and *Streptococcus iniae*, Rb-IL15Ra of *Oplegnathus fasciatus* increased in the trunk kidney, head kidney and spleens [18]. Ma-IL15Rα has important defensive effects on *Flavobacterium columnare* G4, *Ichthyophthirius multifiliis* and *Saprolegnia parasitica* infection of dojo loaches [16]. However, the systemic immune-protective mechanisms of fish IL15Rα during a pathogenic infection are still unclear. In our previous study, Ec-IL15 significantly increased the immune protection of groupers [19]. Compared with the experimental results, both Ec-IL15 and Ec-IL15Rα had a positive protective effect, but it was more obvious in Ec-IL15. Therefore, we reasonably speculated that Ec-IL15Rα cooperated with Ec-IL15 as a constitutive expression pattern to mediate signal transduction in the form of trans-presentation, to better activate the immune-relevant signaling pathways.

The orange-spotted grouper is an important agricultural product in southern China and Southeast Asia [20]. However, in grouper farming, high intensification is usually accompanied by pathogenic disease outbreaks, like vibriosis, which reduce production dramatically [21]. A variety of aquatic and marine environments harbor *Vibrio harveyi*, a kind of Gram-negative opportunistic pathogen [22], which causes serious diseases in all kinds of aquatic animals in China, including *Litopenaeus vannamei* [23], *Larimichthys crocea* [24], *Epinephelus coioides* [25] etc. *V. harveyi* had arisen in many areas of cultured grouper, devastating grouper breeding industries and reducing the production of groupers. Although antibiotics used to control vibriosis in grouper, they also caused severe consequences, including drug resistance and drug residual issues [26]. Understanding the immunomodulatory mechanism of host response to vibriosis in order to find effective treatments for combating vibriosis is urgently needed. In this study, IL15Rα of *E. coioides* was first demonstrated, and the roles of Ec-IL15Rα in regulating immune response against *V. harveyi* infection were also investigated. The present study contributes to a better understanding of the role of Ec-IL15Rα in preventing vibrio infection in groupers.

## 2. Materials and Methods

### 2.1. Fish and Bacterium Preparation

Healthy orange-spotted groupers with body weight 65.0 ± 5.0 g and body length 10.0 ± 2.0 cm were purchased from a grouper breeding base in Dapeng New District, Shenzhen City, Guangdong Province, China. Groupers were cultured in sea water at 26.0 ± 2.0 °C. The sea water parameters were as follows: salinity was 30.0 ± 2.0‰, dissolved oxygen was 5.0 ± 1.0 mg/L, pH was 8.0 ± 0.2, and total alkalinity was 100.0 ± 5.0 mg/L. Groupers were fed about 2~3% of their body weight in feed a day for 2 weeks. Before injection, fish were anesthetized with 100 mg/mL tricaine methane sulfonate (MS-222). 

The strain ZJ0603 of *V. harveyi* (PRJNA165825) was stored in our laboratory. This bacterium was isolated from grouper *Epinephelus coioides*. The bacterium was cultured in TSB at 28 °C overnight, washed in phosphate-buffered saline (PBS), and adjusted to the concentration of 1 × 10^9^ CFU/mL [19].

### 2.2. Open Reading Frame Cloning and Sequence Analysis of IL15Rα

The total RNA was extracted and first-strand cDNA was synthesized from the spleen of healthy groupers using TransZol Up Plus RNA Kit (TransGen Biotech, Beijing, China) and EasyScript^®^ One-Step gDNA Removal and cDNA Synthesis SuperMix (Takara, Beijing, China), respectively. Based on the *E. coioides* transcriptome database, we designed primers for *IL15Rα*, and listed them in Table 1. The method of sequence analysis was consistent with that described by Wang [27].

### 2.3. Temporal Expression of IL15Rα in 11 V. harveyi-Stimulated Tissues and Healthy Tissues

*V. harveyi* (100 μL, with the final concentration of 1 × 10^9^ CFU/mL) was intraperitoneally injected into 18 groupers. The total RNA was extracted and cDNA was amplified using same methods of Section 2.2. The qRT-PCR amplification was performed using PerfectStart^®^ Green qPCR SuperMix (TransGen Biotech, Beijing, China). *V. harveyi*-stimulated tissues including liver, blood, intestine, head kidney, spleen, skin, gills, thymus, muscle, brain and heart were obtained from injected groupers (3 fish per time point) at 0, 6, 12, 24, 48 and 72 h time points, with 0 h as control. Healthy tissues including liver, blood, intestine, head kidney, spleen, skin, gills, thymus, muscle, brain and heart were obtained from healthy groupers. Fold change of *IL15Rα* in tissues was relative to the expression in the heart as a control. Primers are listed in Table 1.

### 2.4. Expression Pattern of IL15Rα in Head Kidney Lymphocytes (HKLs)

HKLs were prepared as previously described with minor modifications [28]. HKLs (10^6^ cell/mL) were added in 48-well plates and divided into 5 groups. Then, 250 μL per well of 4 different pathogenic stimuli, including conA type A IV (ConA, Solarbio, Beijing, China), lipopolysaccharides (LPS, Solarbio, Beijing, China), phytohemagglutinin PHA-P (PHA, Solarbio, Beijing, China) and polyinosinic-polycytidylic acid (poly I:C, shyuanye, Shanghai, China), with a concentration of 10 μg/mL, were mixed with HKLs; PBS was used as the control group. The mixtures were incubated and collected at six time points (0, 3, 6, 9, 12 and 24 h) after the addition, respectively.

### 2.5. Ec-IL15Rα Subcellular Localization Analysis

According to the Lipofectamine™ 3000 Transfection Reagent kit (Thermo fisher scientific, Waltham, MA, USA), the following instructions were followed for transfection experiments. HEK-293T cells were added and cultured in 24-well plates at 37 °C in Dulbecco’s modified Eagle’s medium (with 10% fetal bovine serum and 5% CO_2_) for 12 h. Then, empty pEGFP-N1 and pEGFP-*IL15Rα* were transfected into 293T cells. The cells were washed with PBS, fixed with 4% paraformaldehyde, and stained with DAPI (1 g/mL) 24 h after transfection. Finally, confocal fluorescence microscope (Wetzlar, Germany) was used to observe the cells after they had been rinsed with PBS and mounted with 50% glycerol.

### 2.6. Preparation of Ec-IL15Rα Recombinant Protein

The method of protein expression and purification was consistent with what Wang described [27]. The primers with restriction sites of *EcoR* I and *Xho* I were designed to amplify *IL15Rα* open reading frame (ORF) (Table 1). EcIL15Rα has four conserved cysteine residues in the ectodomain, which is the basis for IL15Rα to form the sushi domain and thus function. The extracellular domain protein gene of *IL15Rα* was ligated with pGEX-4T-1 vector. The recombinant vector was transformed into *Escherichia coli* BL21 (DE3) (TransGen, Beijing, China). The positive clone was cultured in Luria–Bertani (LB) added 100 mg/mL Ampicillin. After OD_600_ reached 0.4–0.6, 0.5 mmol/L, isopropyl-β-D-thiogalactopyranoside (IPTG) was added. The bacteria were induced for 6 h, then collected and washed using PBS. The protein was purified and analyzed by SDS-PAGE and Western blot. 

### 2.7. Ec-IL15Rα Recombinant Protein Induced HKLs Cytokine Production and Proliferation

The Ec-IL15Rα recombinant protein (rEc-IL15Rα) was used to test cytokine production and proliferation of HKLs in vitro. The regulation of rEc-IL15Rα to the expression level changes in 4 cytokines, including pro-inflammatory factors IL1R, IL6R, and IL16, and anti-inflammatory factor IL-10, were detected in HKLs. 500 μL HKLs with the concentration of 2000 cells/μL were added into 24-well plates and divided into 3 groups: GST, PBS and rEc-IL15Rα group. Then, GST, PBS and rEc-IL15Rα with the concentration of 10 μg/mL were added into 24-well plates, respectively. HKLs were collected after 0, 3, 6, 9, 12 and 24 h. RNA was extracted and cDNA was reverse transcribed. β-actin was used to normalize the mRNA level.

An amount of 100 μL HKLs with the concentration of 50 cells/μL were added into 96-well plates and divided into 4 groups: PHA, PBS, GST and rEc-IL15Rα group. Then, 1 μL/per well (final concentration was 5 μg/mL) of PHA, GST, PBS and rEc-IL15Rα were added into 96-well plates, respectively. An amount of 10 μL CCK-8 (Solarbio, Beijing, China) was added to wells and incubated for 4 h. Finally, microplate reader (Thermo fisher scientific, USA) was used to detect OD_450_. 

### 2.8. Survival Rate of Groupers against V. harveyi Infection

A total of 180 fish were divided into 6 groups (30 fish/group), with VH + GST(L), VH + GST(H), VH + Ec-IL15Rα(L) and VH + Ec-IL15Rα(H) as the experimental groups, and the PBS group and VH groups as the negative control (Table 2). To test the function of Ec-IL15Rα in vivo, fish were intraperitoneally injected with 100 μL *V. harveyi* and the survival rate (SR) was detected from 1 to 7 d. Bacterium was re-isolated from the liver of dead fish and incubated on TSA plates. 16s rDNA was used to identify *V. harveyi* infection in fish. The SR was calculated as SR = (surviving fish ÷ 30) × 100%. The experiment was repeated three times.

### 2.9. Drawings and Statistical Analysis

Adobe Photoshop CC (San Jose, CA, USA) and Adobe Illustrator (San Jose, CA, USA) were used to create the drawings and final panels. SPSS 26.0 software with Duncan’s new multiple range test was used to analyze the data using one-way analysis of variance (ANOVA).

## 3. Results

### 3.1. Open reading frame Cloning and Sequence Analysis of IL15Rα

*IL15Rα* contains 720 bp ORF that encodes the putative protein of 239 amino acids, including a 26 amino acid signal peptide domain and a 23 amino acid transmembrane domain. SMART domain architecture analysis predicted that Ec-IL15Rα contains a CCP (sushi) domain that was from 31 to 94 amino acids (Figure 1A). Ec-IL15Rα has predicted the molecular mass of 25.6 kDa, and its theoretical pI is 4.93. Phylogeny analysis revealed that Ec-IL15Rα closely related to *E. lanceolatus* and mainly clustered with teleost (Figure 1B). Multiple sequence alignment indicated that Ec-IL15Rα had four typical conserved cysteine residues, which are the basis for *IL15Rα* to form the CCP (sushi) domain (Figure 1C).

### 3.2. Tissues Expression Pattern of Ec-IL15Rαin V. harveyi-Infected and Healthy Groupers

*Ec-IL15Rα* mRNA levels in tissues of healthy orange-spotted groupers were measured (Figure 2). After *V. harveyi* infection, *Ec-IL15Rα* in head kidney, spleen and gill were significantly up-regulated and up to top at 6 h post-infection (Figure 2D,E,G). *Ec-IL15Rα* expression did not reach their peak until 12 h in the liver, blood, skin, muscles, brain and heart (Figure 2A,B,F,I–K). *Ec-IL15Rα* expression in the thymus reaches its highest level at 6 h, then declines and reaches a maximum again at 48 h (Figure 2H). The *Ec-IL15Rα* expression level in the intestinal continued to decrease for 12 h, then returned to a normal level in 24 h (Figure 2C). In healthy grouper, *Ec-IL15Rα* was most highly expressed in the liver, then in blood, intestine, head kidney, spleen, skin, gill, thymus, muscles, and brain, with heart having the lowest expression level (Figure 2L).

### 3.3. Expression Patterns of Ec-IL15Rα in HKLs Stimulated by Different Pathogenic Stimuli

After being stimulated with ConA, PHA, LPS, and poly I:C in HKLs, the expressed pattern of *Ec-IL15Rα* was observed. The results showed that *Ec-IL15Rα* expression was rapidly maximized at 6 h by ConA, LPS and poly I:C stimulation (Figure 3A,C,D), whereas the expression of *Ec-IL15Rα* did not reach its peak until 9 h after PHA stimulation (Figure 3B).

### 3.4. Subcellular Localization Analysis of Ec-IL15Rα

The subcellular position of Ec-IL15Rα was identified by expressing the pEGFP-Ec-IL15Rα fusion protein in HEK-293T cells and examining it with a fluorescence microscope. The nucleus was stained blue with DAPI, pEGFP-N1 was green fluorescent protein, and was used as the control group. In Figure 4A, the control group was expressed in whole cell and pEGFP-Ec-IL15Rα was mainly expressed in cytoplasm and plasma membrane.

### 3.5. Prokaryotic Expression of Recombinant Protein Ec-IL15Rα (rEc-IL15Rα)

The recombinant vector pGEX-4T-1-Ec-IL15Rα was constructed and induced to produce the recombinant protein rEc-IL15Rα. The molecular weight of rEc-IL15Rα was consistent with the predicted result, with a molecular weight of 25.6 kDa (Figure 4B,C). 

### 3.6. Cytokine Production and Proliferation Induced by Ec-IL15Rα in HKLs

HKLs were incubated with rEc-IL15Rα, then the expressions of cytokines and cytokines receptor including *IL16* and *IL-10*, and *IL1R* and *IL6R* were detected for 3, 6, 9, 12 and 24 h to evaluate the immune protective effect of Ec-IL15Rα. *IL1R* was only significantly up-regulated at 12 h (Figure 5A). *IL6R* significantly increased at 6 h, 12 h and 24 h, with the highest expression level at 6 h. (Figure 5B) The mRNA expression level of *IL10* increased at 3 h and reached to peak at 12 h (Figure 5C). *IL16* maximized at 3 h, and increased at 9 h and 24 h (Figure 5D). The results showed that Ec-IL15Rα could positively stimulate the *IL1R*, *IL6R*, *IL10* and *IL-16* expression. 

The proliferative effect of Ec-IL15Rα on HKLs was detected using the CCK-8 method. The results showed that PBS and GST had no significant effect on the proliferation of HKLs, whereas PHA and Ec-IL15Rα could positively stimulate HKL proliferation (Figure 5E).

### 3.7. Survival Rate of Ec-IL15Rα against V. harveyi Infection

The SR of the VH+Ec-IL15Rα(L) and VH+Ec-IL15Rα(H) groups started decreasing from 2 d post-injection and finally decreased to 10% and 6.67% at 7 d, respectively. The results showed that compared with PBS, VH+GST(L) and VH+GST(H) groups with the SR of 0%, IL15Rα protein showed an increase of 6.67–10% in relative survival rate against *V. harveyi* infection (Figure 5F).

## 4. Discussion

IL15Rα, as a receptor of IL15, was well characterized in avians, mammals and teleosts. However, the functions of orange-spotted grouper IL15Rα are still unclear. In order to interpret the function of IL15Rα, our study identified and described the molecular characterizations of IL15Rα from orange-spotted grouper (Figure 6). We also evaluated the mRNA level of *Ec-IL15Rα* in 11 tissues and under four stimuli, as well as tested the immune-protective effects of Ec-IL15Rα protein in vivo and in vitro. These results provide insight for understanding the function and immune mechanism of IL15Rα in fish against bacterial infection.

The human and mouse IL15R complex is stable, and makes it extremely difficult for the free form of IL15 to be detected in humans and mice [10]. Vice versa, a soluble sushi domain of IL15Rα could bear most of the binding affinity for IL15, to enhance the binding and biological effects (proliferation and protection from apoptosis) [15,29,30]. The cysteine residue serves as the basis of IL15Rα to form the sushi domain, as well as plays an important role in the formation of two disulfide bonds for proper protein folding and IL15-IL15Rα interactions [10,16,31]. Ec-IL15R were comparable to other species in that they had a signal peptide, a sushi domain, and a transmembrane domain [10], all of which were essential for IL15 binding and IL15R function [15,29]. Multiple sequence alignments and the phylogenetic tree showed that Ec-IL15Rα shares about 55% similarity with teleost IL15Rα, about 23% similarity with avian IL15Rα and about 17% similarity with mammal IL15Rα. The analyzed results are consistent with the situation of Ma-IL15Rα [16], Rb-IL15Rα [18] and Rt-IL15Ra [17]. In conclusion, fish IL15Rα had higher similarity and clustered with other teleost IL15Rα and had lower similarity with avians and mammals. In the long evolution from fish to mammals, IL15Rα may have had to evolve in different directions in order to adapt to the changes in different environment. 

*IL15Rα* was proved to be widely expressed in a variety of cell types and tissues, and mainly expressed in immune-relevant tissues that could be activated by immune response to adjust bacterium challenge. *Ec-IL15Rα* was obviously up-regulated within 48 h in tissues after *V. harveyi* infection. The mRNA levels of *Ec-IL15Rα* were obviously positively regulated and maximized at 6 h post-infection in gill, spleen and head kidney, at 12 h in the liver, heart, brain and muscles, at 48 h in thymus, and at 72 h in the intestines. Although different expression patterns were detected in different fish, *IL15Rα* was widely distributed and could certainly be induced by bacterial infection to play roles in immune defense. A similar conclusion was drawn in several studies: that the *IL15Rα* in different tissues can be enhanced after *M. anguillicaudatus* challenge against *F. columnare* [16], and rock bream against *E. tarda* and *S. iniae* [18]. In teleost, *IL15Rα* was highly expressed in the blood, brain, skin and gill of *M. anguillicaudatus* [16], while different tissues (head kidney and trunk kidney) with high expression were detected in *O. fasciatus* [18]. In our study, the *Ec-IL15Rα* was detected in 11 tissues, with the highest levels in the liver and blood. *Ec-IL15Rα* was highly expressed in immune-related tissues in both bacteria-infected and healthy groupers, indicating that *IL15Rα* may play key roles throughout the immune process of the body. Subsequently, HKLs were isolated and stimulated by ConA, PHA, LPS, and poly I:C, respectively. Results revealed that *IL15Rα* was significantly up-regulated by four pathogenic stimuli, which showed that *IL15Rα* could stimulate immune defense. The above results indicate that *Ec-IL15Rα* could be expressed in both immune and non-immune organs to activate immune responses in fish. The same results were shown in *Ec-IL15* [19]. Until now, the specific immune function and mechanism of Ec-IL15Rα remain unclear. Therefore, we reasonable speculate that *Ec-IL15Rα* cooperated with *Ec-IL15* as a constitutive expression pattern to mediate signal transduction in the form of trans-presentation, to better activate the immune-relevant signaling pathways.

In this study, IL15Rα contains 720 bp ORF that encodes the putative protein of 239 amino acids, including a 26-amino-acid signal peptide domain and a 23-amino-acid transmembrane domain (Figure 1). In HEK-293T cell line, pEGFP-IL15Rα is mainly distributed in the cytoplasm and plasma membrane. The result was consistent with the prediction that Ec-IL15Rα protein has a transmembrane domain. Likewise, in a previous study, the IL15 of orange-spotted grouper was mainly distributed in the same situation [19], which is consistent with the hypothesis that IL15Rα and IL15 combine to mediate signal transduction in the form of trans-presentation and are characterized by constitutive expression in the body. SDS-PAGE and Western blot of rEc-IL15Rα results (Figure 4B,C) were all consistent with the predicted molecular mass of 25.6 kD (Figure 1).

In order to investigate whether Ec-IL15Rα proteins can activate downstream signaling molecules in HKLs, the function of Ec-IL15Rα on the cytokine production and proliferation of HKLs was detected in vitro. As the results suggest, cytokines and cytokine receptors including *IL1R*, *IL6R*, *IL10* and *IL16* could be activated by Ec-IL15Rα. IL1 and IL10, as inflammatory cytokines, could provide innate inflammation that is required for nonspecific host defense as well as acquired immunity [32]. IL6R as an IL6 receptor can bind to IL6 to form the IL-6/IL-6R complex, leading to a wave of receptor, JAK, and STAT phosphorylation, and culminating in the nuclear import of phosphorylated STAT dimers, predominantly STAT3, that activate transcription, leading to growth and cellular differentiation [33]. As a multi-effector cytokine, IL16 can not only cause the migration of CD4^+^ T lymphocytes, monocytes and eosinophils, but also initiate CD4^+^ T cells and induce T lymphocytes to express IL2 receptor [34]. In addition, IL16 negatively regulates IL10 expression, positively regulates IL1Rα, IL6 and other pro-inflammatory cytokines, and improves the phagocytosis ability of macrophages [34]. As a result, the injection of rEc-IL15R strongly activated cytokines and cytokine receptors in HKLs cells, with the maximum expression of IL1R and IL10 occurring at 12 h after incubation and that of IL6R and IL16 occurring at 6 h after incubation. However, these results only showed that cytokines could be activated at mRNA levels; the specific mechanism of how Ec-IL15Rα regulate cytokine protein expression in grouper needs further study. Proliferation could be used as one of the indicators to determine the immune level of the cell. Our results showed that Ec-IL15Rα could positively stimulate the HKL proliferation for 4 h post-incubation. Taken together, Ec-IL15Rα not only significantly activated downstream signaling molecules, but also stimulated the proliferation of HKL. Therefore, Ec-IL15Rα could stimulate immune response in vitro.

In vivo, most reports showed that IL15 or IL15/IL15Rα complex could effectively improve immune protection to the host, including female BALB/c mice [35], *Danio rerio* [36], and female C57Bl/6 mice [37]. IL15 or IL15/IL15Rα complex induces robust proliferation of NK cells, NK T cells, memory CD8^+^ T cells, etc., promoting immune response and anti-infection [7,38,39,40,41]. In this study, we also examined whether the rEc-IL15Rα recombinant protein has similar immune protective function of groupers in vivo. We found the high or low concentration rEc-IL15Rα injection had an SR of 10% and 6.67%, respectively, in groupers. Interestingly, Wu et al. showed that the SR of high or low concentration rEc-IL15 were 53.3% and 80%, respectively, in the same groupers [19]. Compared with Ec-IL15, Ec-IL15Rα only increased 6.67–10% SR in groupers against *V. harveyi* infection. Suggesting that Ec-IL15Rα may play roles in enhancing the binding and assisting Ec-IL15, to co-express and operate together against pathogen infection.

In conclusion, IL15Rα from orange-spotted grouper was cloned, and the amino acid sequence and structure of Ec-IL15Rα showed high similarity with other teleost species. *Ec-IL15Rα* were expressed in all 11 tested tissues, with the highest expression in liver. *Ec-IL15Rα* could be stimulated by ConA, PHA, LPS and poly I:C stimuli, and take part in the immune response against infective *V. harveyi*. A subcellular localization study showed that Ec-IL15Rα is distributed in both cytoplasm and cell membrane. Although Ec-IL15Rα only improved 6.67%-10% of SR in vivo, but it could stimulate activate *IL1R*, *IL6R*, *IL10* and *IL16* expression and proliferation of HKL in vitro. This study establishes the groundwork for future research into the mechanism through which IL15R defends fish from infections.

## Figures and Tables

**Figure 1 animals-13-03641-f001:**
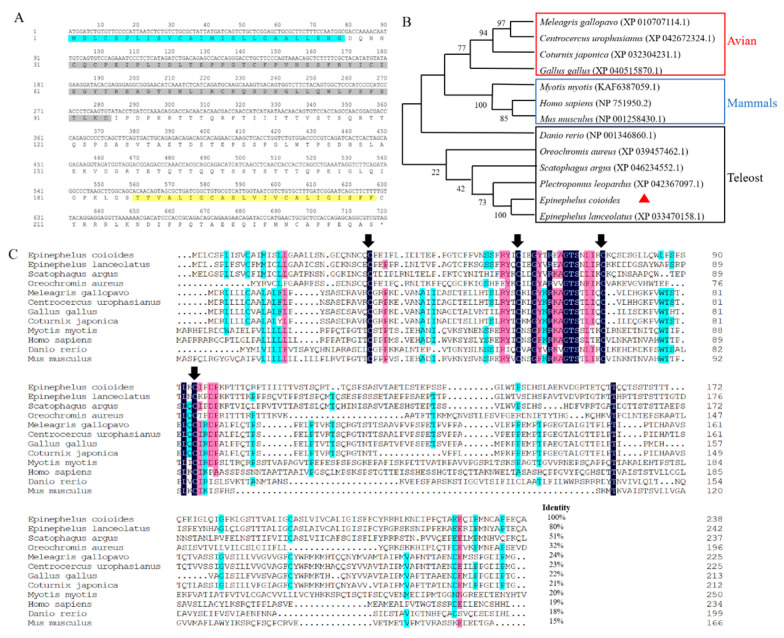
Identification and analysis of *Ec-IL15Rα*. (**A**) Open reading frame and deduced amino acid sequences of *Ec-IL15Rα*. ORF is capitalized, asterisk indicates the stop codon, putative signal peptide domain is blue-shaded, conserved domains are gray-shaded and putative transmembrane domain is yellow-shaded. (**B**) Phylogenic tree construction of Ec-IL15Rα. Phylogenic analysis was conducted by N-J method in MEGA 7.0. Avian: *Meleagris gallopavo* XP 010707114.1, *Centrocercus urophasianus* XP 042672324.1, *Coturnix japonica* XP 032304231.1; Mammals: *Myotis myotis* KAF6387059.1, *Homo sapiens* NP 751950.2, *Mus musculus* NP 001258430.1; Teleost: *Danio rerio* NP 001346860.1, *Oreochromis aureus* XP 039457462.1, *Scatophagus argus* XP 046234552.1, *Plectropomus leopardus* XP 042367097.1, *Epinephelus lanceolatus* XP 033470158.1, *Epinephelus coioides*. The red triangle represents this experiment studied species. (**C**) Multiple sequence alignments of Ec-IL15Rα from different species, including: *Epinephelus coioides*, *Epinephelus lanceolatus*, *Scatophagus argus*, *Oreochromis aureus*, *Meleagris gallopavo*, *Centrocercus urophasianus*, *Gallus*, *Coturnix japonica*, *Myotis*, *Homo sapiens*, *Danio rerio*, *Mus musculus*. The 4 conserved cysteine residues are marked in black arrows.

**Figure 2 animals-13-03641-f002:**
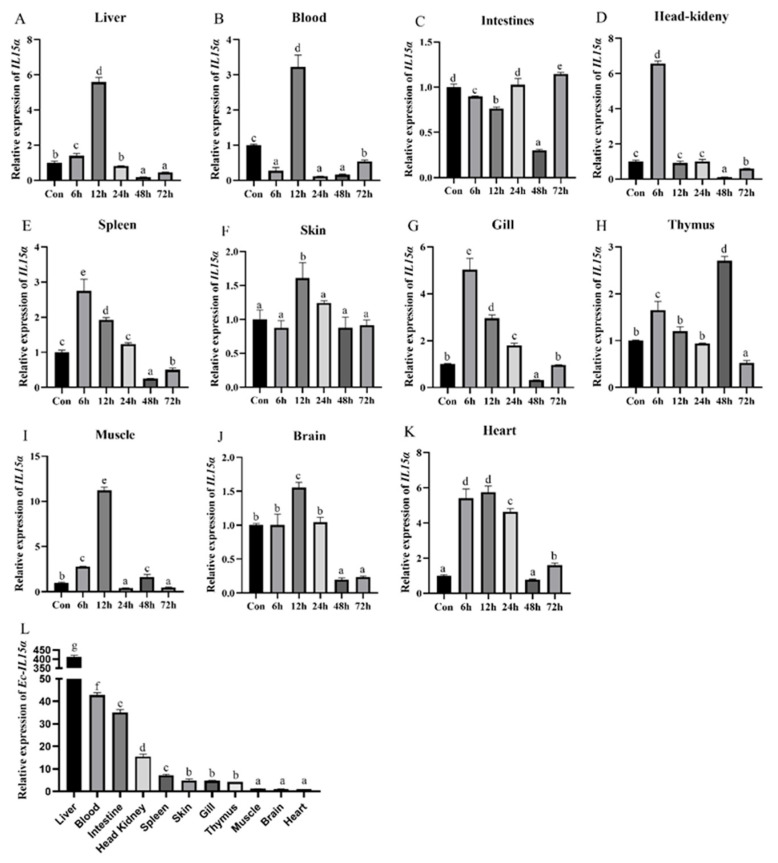
Expression patterns of *Ec-IL15Rα* in 11 different tissues (**A**–**K**) at 0, 6, 12, 24, 48, 72 h time points after *V. harveyi* injection and (**L**) in healthy orange-spotted groupers, respectively, were detected by qRT-PCR. (**A**–**K**) The expression level of *Ec-IL15Rα* at 0 h was set as control. (**L**) Fold change of *IL15Rα* in tissues was relative to the expression in heart (as control). All values are the mean ± SE, *n* = 3. Significant differences (*p* < 0.05) are marked by different letters (a–g).

**Figure 3 animals-13-03641-f003:**
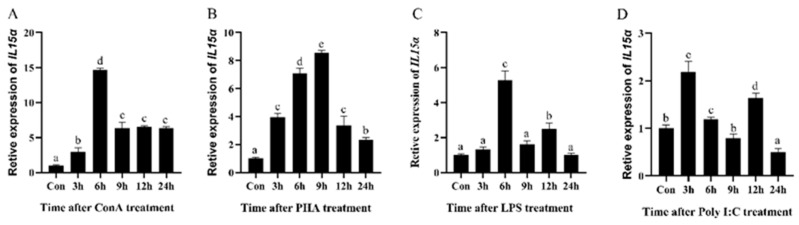
Expression patterns of *Ec-IL15Rα* in HKLs stimulated by 4 different stimuli, (**A**)—ConA (conA type A IV), (**B**)—PHA (phytohemagglutinin PHA-P), (**C**)—LPS (lipo-polysaccharides), and (**D**)—poly I:C (polyinosinic-polycytidylic acid), were detected by qRT-PCR at 6 different time points. All values are the mean ± SE, *n* = 3. Significant differences (*p* < 0.05) are marked by different letters (a–e).

**Figure 4 animals-13-03641-f004:**
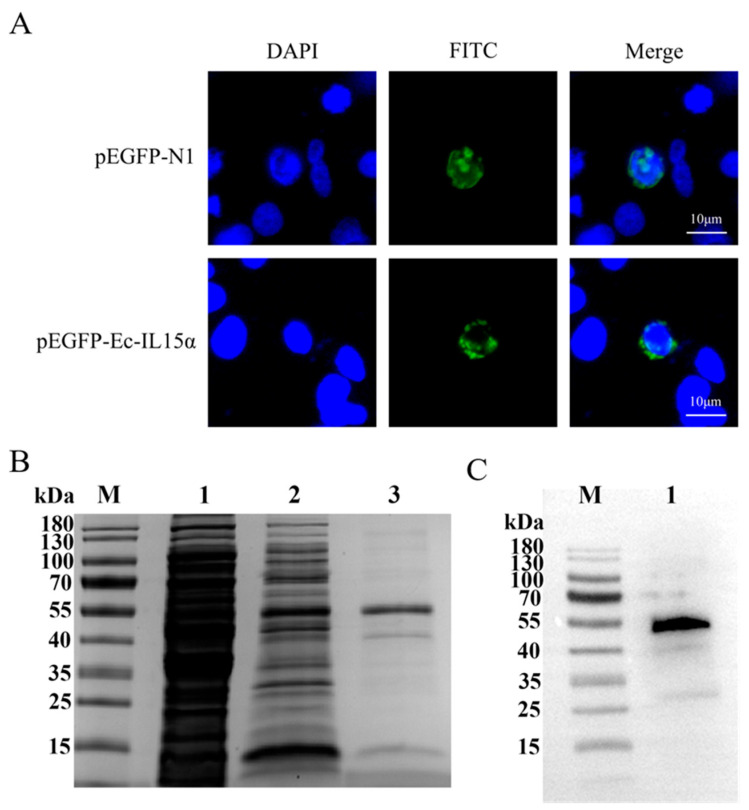
Subcellular localization of Ec-IL15Rα in HEK-293T cells by fluorescence microscopy, SDS-PAGE and Western blot of rEc-IL15Rα. (**A**) The nucleus was stained with 4′,6-dimaidino-2-pheny-lindole (DAPI). Distribution of Ec-IL15Rα is mainly in the cytoplasm and plasma membrane. (**B**) SDS-PAGE analysis. Lane M, markers; lane 1, bacteria before IPTG induction; lane 2, bacteria after IPTG induction; lane 3, purified rEc-IL15Rα. (**C**) Western Blot. M, markers; lane 1, Western blot result of rEc-IL15R.

**Figure 5 animals-13-03641-f005:**
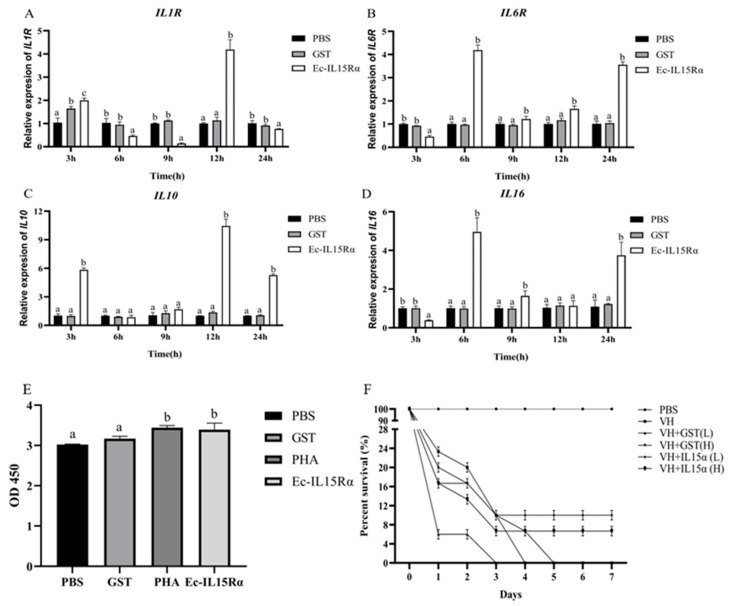
Effects of Ec-IL15Rα on the cytokine expression, proliferation on HKLs, and the survival rate of orange-spotted groupers infected by *V. harveyi*. (**A**–**D**) Expression level changes of 4 cytokines including pro-inflammatory factors: *IL1R*, *IL6R*, *IL16*, and anti-inflammatory factor: *IL-10*, in HKLs were detected after 0, 3, 6, 9, 12, 24 h. The mRNA level of each cytokine gene was normalized to that of β-actin. (**E**) PHA, PBS, GST and rEc-IL15Rα were added to test HKL proliferation. (**F**) Daily fish deaths were recorded after infection with *n* = 30 for each group. All values are the mean ± SE, *n* = 3. Significant differences (*p* < 0.05) are marked by different letters (a–b).

**Figure 6 animals-13-03641-f006:**
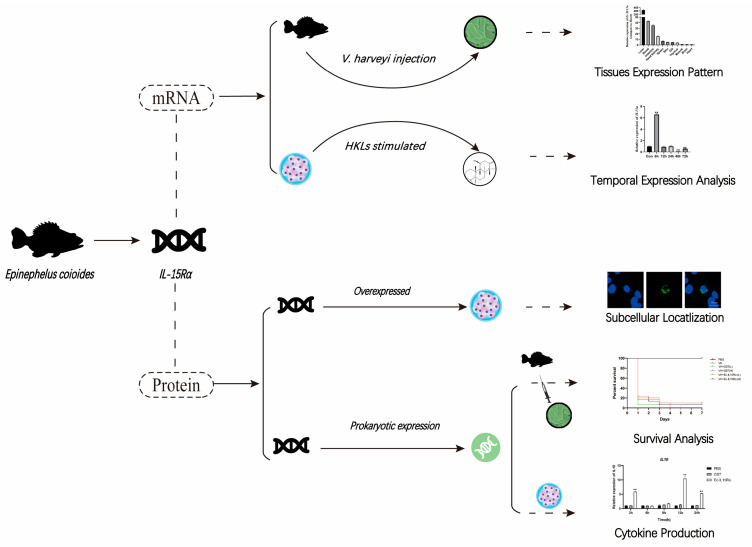
Graphical abstract of Ec-IL15Rα of this experiment scheme. At mRNA level, first graph tissues expression pattern of IL15Rα were tested in both healthy and *V. harveyi* infected groupers. second graph temporal IL15Rα expression level was analysis after HKL cells stimulated. At protein level, (DAPI image) the subcellular localization of Ec-IL15Ra in HEK-293T cells was analyzed. Cytokine expression level and proliferation of HKL cells, as well as the survival of groupers, were tested after *V. harveyi* infection, to better evaluated the immune effect of Ec-IL15Ra.

**Table 1 animals-13-03641-t001:** Primers used in this study.

Primers	Sequence (5′–3′)	Purpose
IL15Rα-ORF-F	ATGGATCTGTGTTCCCCATTAATCT	ORF cloning
IL15Rα-ORF-R	CTACGACGCCTGCTCTGGTGGA	
16s rDNA-F	TTGCGAGAGTGAGCGAATCC	*V. harveyi* identification [18]
16s rDNA-R	ATGGTGTGACGGGCGGTGTG	
IL15Rα-LOC-F	CGGGGTACCGCCACCATGGATCTGTGTTCCCCA	Subcellular localization
IL15Rα-LOC-R	TTTGGGCCCGCGACGCCTGCTCTGGTGG	
IL15Rα-PRO-F	CCGGAATTCATGGATCTGTGTTCCCCA	Protein expression
IL15Rα-PRO-R	CCGCTCGAGCTAGCTGCCAAGCTTAGGG	
IL1R-F	TTGGCCAGTCAGATGGTTCC	qPCR
IL1R-R	GACTGTGGGTGAATGCCGTA	
IL6R-F	GAGAACACAAGCCTCGGACA	
IL6R-R	GACGCCCCTCTCCTCTCTAA	
IL10-F	TTAAGGCCATGGGTGACCTG	
IL10-R	CAGCAAGCAGCAACAACACT	
IL16-F	TGAGCTCAACCACCATCACC	
IL16-R	TTTCCGTCAGACTTTGGCGA	
β-actin-F	TGCTGTCCCTGTATGCCTCT	
β-actin-R	CCTTGATGTCACGCACGAT	

IL15Rα: interleukin-15 receptor α; 16s rDNA: *V. harveyi* identification; IL1R: interleukin 1 receptor; IL6R: interleukin 6 receptor; IL10: interleukin 10 receptor; IL16: interleukin 16 receptor; β-actin: as control.

**Table 2 animals-13-03641-t002:** Groups of SR experiment.

Groups	Injection	Dose
PBS group	intraperitoneal injection	100 μL/fish
VH group	intraperitoneal injection	100 μL/fish (1 × 10^8^ CFU *Vibrio harveyi*)
VH+GST(L)	intraperitoneal injection	100 μL/fish (10 μg GST protein and 1 × 10^8^ CFU bacterial mixture)
GST(H)	intraperitoneal injection	100 μL/fish (50 μg GST protein and 1 × 10^8^ CFU bacterial mixture)
VH+Ec-IL15Rα(L)	intraperitoneal injection	100 μL/fish (10 μg rEc-IL15Rα and 1 × 10^8^ CFU bacterial mixture)
VH+Ec-IL15Rα(H)	intraperitoneal injection	100 μL/fish (50 μg rEc-IL15Rα and 1 × 10^8^ CFU bacterial mixture)

PBS—Phosphate-buffered saline; VH—*Vibrio harveyi*; GST—glutathione-S-transferase; Ec-IL15Rα—α chain of the IL15 receptor of *Epinephelus coioides*; rEc-IL15Rα—Ec-IL15Rα recombinant protein.

## Data Availability

The data that support the findings of this study are available from the corresponding author upon reasonable request.

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
