# Peer review of "Molecular and Functional Characterization of α Chain of Interleukin-15 Receptor (IL-15Rα) in Orange-Spotted Grouper (Epinephelus coioides) in Response to Vibrio harveyi Challenge"

_animals, 2023, doi:10.3390/ani13233641_

Round 1
Reviewer 1 Report
Comments and Suggestions for Authors
The manuscript entitled “Molecular and functional characterization of α chain of interleukin-15 receptor IL-15Rα in Orange-spotted grouper (Epinephelus coioides) in response to Vibrio harveyi challenge” aims to identify the α chain of IL15 receptor of Epinephelus coioides and determine the immune response functions of the fish after Vibrio harveyi challenge. The introduction is streamlined and clear, the experimental protocol should be improved and the data obtained presented clearly and adequately discussed. The manuscript may be considered with a revision as detailed below.
General comments:
· All scientific names should be present in Italics.
· Check all grammatical errors in the whole manuscript.
Abstract:
· For better understanding you should provide a graphical abstract.
Introduction:
· Elaborate the research gaps. Mention the hypothesis of this study.
Materials and methods:
· Line no 76: Mention the length of the fish also.
· Line no 78: It is necessary to mention the other important water quality parameters like salinity, DO, pH, and alkalinity.
· Line no 81: The strain ZJ0603 of V. harveyi was stored in our laboratory………. Was the strain previously used in another study? Mention the accession number of the bacterium.
· Line no 83-84: adjusted to the concentration of 1×109 CFU/mL………. How the CFU was calculated and adjusted?
· Line no 86-87: The first-strand cDNA was synthesized from total RNA extracted from healthy grouper………… The RNA isolation protocol should have been written previously. You can rearrange your points.
· Line no 98: 2.4. RNA Isolation and cDNA Synthesis ………… RNA was isolated from which tissue? Briefly write the RNA isolation and cDNA synthesis protocol.
· Line no 106-107: V. harveyi (100 μL, with the final concentration of 1×109 CFU/mL) was intraperitoneally injected into 18 groupers……….. Was there any control group? Which fish were used as control group? Was the fish of control group injected with PBS (100 μL)?
· Line no 131: 2.6.2. Survival rate of groupers against V. harveyi infection ………… The experiment should be set up in triplicate.
Results:
· Figure 3, 4, 5 and 8: The asterisk * should be changed to a, b, c and d to understand the significant changes between different group.
· Line no 229: 3.7 Survival rate of Ec-IL15Rα against V. harveyi infection ……….. As the experiment should be set up in triplicate, hence the results should be presented as mean ± SE. The error bar should be also mentioned in Figure 7.
Discussion:
· Try to correlate the results obtained with each other
Comments on the Quality of English LanguageEnglish improvement is required
Author Response
All scientific names should be present in Italics.
Answer: Thank you for the advices. Scientific names were presented in Italics or listed in table 1.
Check all grammatical errors in the whole manuscript. Answer: Thank you for your comment, we have checked the full text and revised it. For better understanding you should provide a graphical abstract.
Answer: Thank you for the advices. “Figure 6 Graphical abstract of Ec-IL15Rα in experiment scheme” was added.
Elaborate the research gaps. Mention the hypothesis of this study.
Answer: Thank you for the advices. The hypothesis “In our previous study, Ec-IL15 significantly increased immune protection of groupers [26]. Compared with the experimental results, both of Ec-IL15 and Ec-IL15Rα had the positive protective effect, but the more obvious of Ec-IL15. Therefore, we reasonable speculated that Ec-IL15Rα cooperated with Ec-IL15 as constitutive expression pattern, to mediate signal transduction in the form of trans-presentation, to better activating the immune relevant signaling pathways.” was added in the introduction.
Line no 76: Mention the length of the fish also.
Answer: Thank you for the advices, body length 10.0 ± 2.0 cm was added in the manuscript.
Line no 78: It is necessary to mention the other important water quality parameters like salinity, DO, pH, and alkalinity.
Answer: The sea water parameters were as follow: salinity was 30.0 ± 2.0 ‰ï¼Œdissolved oxygen in sea water was 5.0 ± 1.0 mg/L, pH was 8.0 ± 0.2, total alkalinity of seawater was 100.0 ± 5.0 mg/L
Line no 81: The strain ZJ0603 of V. harveyi was stored in our laboratory………. Was the strain previously used in another study? Mention the accession number of the bacterium.
Answer: Thank you for your advices. The accession number (PRJNA165825) was added in the manuscript. “The strain ZJ0603 of V. harveyi (PRJNA165825) was stored in our laboratory”.
Line no 83-84: adjusted to the concentration of 1×109 CFU/mL………. How the CFU was calculated and adjusted?
Answer: Thanks for your advices. We had calculated the growth curve before. The concentration of Vibrio harveyi was calculated and adjusted to OD600=1.0 as about 1×109 CFU/ml.
Line no 86-87: The first-strand cDNA was synthesized from total RNA extracted from healthy grouper………… The RNA isolation protocol should have been written previously. You can rearrange your points.
Answer: Thanks for your advices. “The total RNA was extracted and first-strand cDNA was synthesized from head kidney of healthy grouper using TransZol Up Plus RNA Kit (TransGen Biotech, Beijing, China) and EasyScript® One-Step gDNA Removal and cDNA Synthesis SuperMix (Takara, Beijing, China), respectively” was added in the manuscript.
Line no 98: 2.4. RNA Isolation and cDNA Synthesis ………… RNA was isolated from which tissue? Briefly write the RNA isolation and cDNA synthesis protocol.
Answer: Thanks for your advices. “The total RNA was extracted and cDNA was amplified from 11 tissues: liver, blood, intestine, head kidney, spleen, skin, gill, thymus, muscles, brain and heart, using same ways of 2.2”.
Line no 106-107: V. harveyi (100 μL, with the final concentration of 1×109 CFU/mL) was intraperitoneally injected into 18 groupers……….. Was there any control group? Which fish were used as control group? Was the fish of control group injected with PBS (100 μL)?
Answer: Thanks for the interesting question. We used the fish that 0 h intraperitoneally injected with V. harveyi (100 μL) as control group, did not use the fish that intraperitoneally injected with PBS as control group.
Line no 131: 2.6.2. Survival rate of groupers against V. harveyi infection ………… The experiment should be set up in triplicate.
Answer: Thanks for your advices. This experiment repeat for three times.
Figure 3, 4, 5 and 8: The asterisk * should be changed to a, b, c and d to understand the significant changes between different group.
Answer: Thanks for your advices. We had changed to the form of “a, b, c and d” in Figure.
Line no 229: 3.7 Survival rate of Ec-IL15Rα against V. harveyi infection ……….. As the experiment should be set up in triplicate, hence the results should be presented as mean ± SE. The error bar should be also mentioned in Figure 7.
Answer: Thanks for your advices. We had added error bar in figure.
Discussion: Try to correlate the results obtained with each other
Answer: Thanks for your advices. The discussion part was revised, to let contents more interlinked. The revised contents were highlight.

Reviewer 2 Report
Comments and Suggestions for Authors
An attempt to clone il-15 receptor alpha chain in grouper. The study was well-designed and I recommend some minor revisions:
1- Statistics must be better explained. Did the authors considered normality and variance homogeneity before running ANOVA?
2- The authors must provide evidence for suitability of b-actin as the housekeeping gene.
3- In the figure 3, the unit of vertical axis is not clear. Does this mean "fold change relative to the expression in heart". The authors must clearly describe the calculation procedure in the methods.
Author Response
1- Statistics must be better explained. Did the authors considered normality and variance homogeneity before running ANOVA?
Answer: Thanks for your question. We used SPSS 26.0 software with Duncan's new multiple range test was used to analyze the data using one-way analysis of variance (ANOVA). In figures, the significantly different between groups was changed from “asterisk *” to the form of “a, b, c, d and e” to understand the significant changes between different group.
2- The authors must provide evidence for suitability of b-actin as the housekeeping gene.
Answer: Thanks for your question. We refer β-actin to several article, therefore the β-actin was suitable as housekeeping gene:
<Mo, Z. Q., Wang, J. L., Han, R., Han, Q., Li, Y. W., Sun, H. Y., Luo, X. C., & Dan, X. M. (2018). Identification and functional analysis of grouper (Epinephelus coioides) B-cell linker protein BLNK. Fish & shellfish immunology, 81, 399–407. https://doi.org/10.1016/j.fsi.2018.07.046.>
< Wu, F., Wang, Z., Yang, G., Jian, J., & Lu, Y. (2022). Molecular characterization and expression analysis of interleukin-15 (IL-15) genes in orange-spotted grouper (Epinephelus coioides) in response to Vibrio harveyi challenge. Fish & shellfish immunology, 128, 327–334. https://doi.org/10.1016/j.fsi.2022.08.003>.
3- In the figure 3, the unit of vertical axis is not clear. Does this mean "fold change relative to the expression in heart". The authors must clearly describe the calculation procedure in the methods.
Answer: Thanks for your advices. “Fold change of IL15Rα in tissues was relative to the expression in heart (as control)” was added in the figure.

Reviewer 3 Report
Comments and Suggestions for Authors
This is an interesting study on IL-15 Ra in orange-spotted grouper. However, there are some concerns, which should be addressed.
Abstract
The sentence "In vivo challenge experiments revealed a certain improvement in survival rates" should be modified according to findings.
Introduction
The authors should clarify *trans-presentation" l. 39, and specify kidney, l. 55: head kidney? A detailed Aim of the study should include the rationale for the different experimental steps, i.e. in vitro, in vivo, bacteria, HEK-293T, etc. to better guide through the manuscript.
Material and methods
The HEK-293T cells should be introduced and the rationale that data on sub-cellular analysis in these cells can be transferred to fish, e.g. by previous research, and should be discussed.
Expressions were calculated using b-actin as reference gene exclusively. Recalculation using another established reference gene or report/citation of experience with other reference genes in this experimental setting is recommended.
Company information have to be accomplished (city, country etc.)
Results
Figure 2: Localization of pEGFP-N1 should be explained as compared to Ec-IL15a, e.g. localization, control etc.
Figure 3: typo (Spleen), Figure 4: could be combined with Fig. 3. It should be clarified in the figure legend that Fig. 3 is without bacteria. Fig 4 should be arranged in analogy to the order of tissues in Fig. 3. At present, there is no clear systematic arrangement of the graphs.
Generally, in vivo and in vitro experiments should be set in a certain order. Fig. 5 (prokaryotic expression of recombinant protein) could be combined with another Figure, e.g. subcellular localization?
At present, the manuscript would be easier to comprehend when the survival rates after infection would be combined with other in vivo infection experiments. Similarly, the authors could consider to combine all in vitro experiments using HKLs together.
Discussion
After restructure of the results sections, the discussion should be restructured accordingly and discuss the respective results, which should be briefly summarised at the beginning of each section. Both, introduction and discussion might benefit from a recent review: 2023 Apr:141:104645. doi: 10.1016/j.dci.2023.104645.
Comments on the Quality of English LanguageGenerally, the manuscript is difficult to read. English editing is recommended. Further, abbreviations are not always introduced at first appearance, some are not explained. This should be thoroughly revised, e.g., bei including abbreviations in the figure and Table legends.
It is recommended to use species names in analogy, i.e. not mix such as mice, zebrafish and Homo sapiens (l. 308).
Author Response
The sentence "In vivo challenge experiments revealed a certain improvement in survival rates" should be modified according to findings.
Answer: Thanks for your advices. We had modified the sentence to “In vivo challenge experiments revealed that IL15Rα protein increased 6.67%-10% survival protection rate after V. harveyi infection”.
The authors should clarify *trans-presentation" l. 39, and specify kidney, l. 55: head kidney? A detailed Aim of the study should include the rationale for the different experimental steps, i.e. in vitro, in vivo, bacteria, HEK-293T, etc. to better guide through the manuscript.
Answer: Thanks for your advices. trans-presentation revised to “Trans-presentation is a mechanism of cytokine delivery that has been describe, that is IL15R and IL-15 encounter each other in the endoplasmic reticulum (ER) and are transported together to the cell surface where the cell surface complex can stimulate neighboring cells through the IL-15Rβ/γC”. Kidney revised to “trunk kidney, head kidney”. The experimental steps were revised in material and results parts. The discussion part was revised, to let contents more interlinked. The revised contents were highlight.
The HEK-293T cells should be introduced and the rationale that data on sub-cellular analysis in these cells can be transferred to fish, e.g. by previous research, and should be discussed.
Answer: Thanks for your advices. The sentences were added in the discussion. “In HEK-293T cell line, pEGFP-IL15Rα is mainly distributed in the cytoplasm and plasma membrane. Likewise, in previous study, IL15 of orange-spotted grouper is mainly distributed in the same situation [26], which is consistent with the hypothesis opinion that IL15Rα and IL15 combine to mediate signal transduction in the form of trans-presentation and are characterized by constitutive expression in the body”. This HEK-293T cells were widely used in our laboratory for subcellular analysis of protein in fish. The references were listed below.
<Wang, Z., Xie, C., Li, Y., Cai, J., Tang, J., Jian, J., Kwok, K. W., & Lu, Y. (2020). Molecular characterization and expression of CD48 in Nile tilapia (Oreochromis niloticus) in response to different stimulus. Fish & shellfish immunology, 97, 515–522. https://doi.org/10.1016/j.fsi.2019.12.034>
<Wang, Z., Xie, C., Li, Y., Cai, J., Tang, J., Jian, J., Kwok, K. W., & Lu, Y. (2020). Molecular characterization and expression of CD48 in Nile tilapia (Oreochromis niloticus) in response to different stimulus. Fish & shellfish immunology, 97, 515–522. https://doi.org/10.1016/j.fsi.2019.12.034>
<Xie, C., Wang, Z., Li, Y., Wu, F., Lu, Y., Xia, H., Tang, J., Jian, J., & Kwok, K. W. (2021). Conservation of structural and interactional features of CD226 and Necl5 molecules from Nile tilapia (Oreochromis niloticus). Fish & shellfish immunology, 116, 74–83. https://doi.org/10.1016/j.fsi.2021.05.014>
Expressions were calculated using b-actin as reference gene exclusively. Recalculation using another established reference gene or report/citation of experience with other reference genes in this experimental setting is recommended.
Answer: Thanks for your question. We refer β-actin to several article, therefore the β-actin was suitable as housekeeping gene:
<Mo, Z. Q., Wang, J. L., Han, R., Han, Q., Li, Y. W., Sun, H. Y., Luo, X. C., & Dan, X. M. (2018). Identification and functional analysis of grouper (Epinephelus coioides) B-cell linker protein BLNK. Fish & shellfish immunology, 81, 399–407. https://doi.org/10.1016/j.fsi.2018.07.046.>
< Wu, F., Wang, Z., Yang, G., Jian, J., & Lu, Y. (2022). Molecular characterization and expression analysis of interleukin-15 (IL-15) genes in orange-spotted grouper (Epinephelus coioides) in response to Vibrio harveyi challenge. Fish & shellfish immunology, 128, 327–334. https://doi.org/10.1016/j.fsi.2022.08.003>.
Company information have to be accomplished (city, country etc.)
Answer: Thanks for your advice. We had added the company information in the manuscript. “Then, 250 μL per well of 4 different pathogenic stimuli, including conA type A IV (ConA, Solarbio, Beijing, Chian), lipopolysaccharides (LPS, Solarbio, Beijing, Chian), phytohemagglutinin PHA-P (PHA, Solarbio, Beijing, Chian) and polyinosinic-polycytidylic acid (poly I:C, shyuanye, shanghai, China)”.
Figure 2: Localization of pEGFP-N1 should be explained as compared to Ec-IL15a, e.g. localization, control etc.
Answer: The sentences were added in 3.5. “The nucleus was stained blue with DAPI, pEGFP-N1 was green fluorescent protein, and was used as the control group. In Figure 4A, the control group was expressed in whole cell and pEGFP-Ec-IL15Rα was mainly expressed in cytoplasm and plasma membrane.”
Figure 3: typo (Spleen), Figure 4: could be combined with Fig. 3. It should be clarified in the figure legend that Fig. 3 is without bacteria. Fig 4 should be arranged in analogy to the order of tissues in Fig. 3. At present, there is no clear systematic arrangement of the graphs.
Answer: Thanks for your advice. We have combined the figure together.
Generally, in vivo and in vitro experiments should be set in a certain order. Fig. 5 (prokaryotic expression of recombinant protein) could be combined with another Figure, e.g. subcellular localization?
Answer: Thanks for your advice. We have combined the figure of “subcellular localization of Ec-IL15Rα in HEK-293T cells by fluorescence microscopy, SDS-PAGE and Western blot of rEc-IL15Rα” together to better analysis the function of Ec-IL15Rα.
At present, the manuscript would be easier to comprehend when the survival rates after infection would be combined with other in vivo infection experiments. Similarly, the authors could consider to combine all in vitro experiments using HKLs together.
Answer: Thanks for your advice. We have combined the figure of “Effects of Ec-IL15Rα on the cytokines expression, proliferation on HKLs, and the survival rate of orange-spotted groupers infected by V. harveyi.” together to better analysis the function of Ec-IL15Rα.
After restructure of the results sections, the discussion should be restructured accordingly and discuss the respective results, which should be briefly summarised at the beginning of each section. Both, introduction and discussion might benefit from a recent review: 2023 Apr:141:104645. doi: 10.1016/j.dci.2023.104645.
Answer: Thanks for your advice. The results sections, the discussion and introduction had made adjustment. The revises were highlight in the manuscript.
Generally, the manuscript is difficult to read. English editing is recommended. Further, abbreviations are not always introduced at first appearance, some are not explained. This should be thoroughly revised, e.g., bei including abbreviations in the figure and Table legends.
Answer: Thanks for your advice. More details were added in both figure and table legends.
It is recommended to use species names in analogy, i.e. not mix such as mice, zebrafish and Homo sapiens (l. 308).
Answer: Thanks for your advice. The species names changed to “female BALB/c mice [32], Danio rerio [33], female C57Bl/6 mice [34”

Round 2
Reviewer 1 Report
Comments and Suggestions for Authors
Paper can be accepted for publication